# Effects of Dietary Alkyl Polyglycoside Supplementation on Lactation Performance, Blood Parameters and Nutrient Digestibility in Dairy Cows

**DOI:** 10.3390/ani9080549

**Published:** 2019-08-13

**Authors:** Xiaoli Zhang, Chunyu Jiang, Qinghua Gao, Duanqin Wu, Shaoxun Tang, Zhiliang Tan, Xuefeng Han

**Affiliations:** 1Key Laboratory for Agro-Ecological Processes in Subtropical Region, National Engineering Laboratory for Pollution Control and Waste Utilization in Livestock and Poultry Production, South Central Experimental Station of Animal Nutrition and Feed Science in the Ministry of Agriculture, Institute of Subtropical Agriculture, The Chinese Academy of Sciences, Changsha, Hunan 410125, China; 2College of Advanced Agricultural Sciences, University of the Chinese Academy of Sciences, Beijing 100049, China; 3Key Laboratory of Animal Husbandry Science and Technology of Xinjiang Production and Construction Corps, College of Animal Science, Tarim University, Alar, Xinjiang 843300, China; 4Hunan Co-Innovation Center of Animal Production Safety, CICAPS, Changsha, Hunan 410128, China; 5Hunan Co-Innovation Center for Utilization of Botanical Functional Ingredients, Changsha 410128, China

**Keywords:** APG, lactation performance, blood metabolites, digestibility, dairy cow

## Abstract

**Simple Summary:**

This project investigated the effects of alkyl polyglycoside (APG), a non-ionic surfactant, on lactation performance, blood metabolites and nutrient digestibility of lactating dairy cows, and found that the addition of APG at doses up to 22 mL/kg of pelleted concentrate (around 9 mL/kg of diet or 100 mL/day/head) had positive effects on milk quality in dairy cows.

**Abstract:**

This study evaluated the effects of alkyl polyglycoside (APG), which is a non-ionic surfactant, on lactation performance, nutrient digestibility and blood metabolites in dairy cows. Twenty dairy cows were randomly divided into four groups and fed a basal diet that included pelleted concentrate, distillers grains, and fresh limpograss. The four treatments included 0, 5.5, 11 and 22 mL APG per kg of pelleted concentrate on a dry matter basis; treatments were defined as APG0, APG5.5, APG11, and APG22, respectively. Dry matter intake was not affected by APG supplementation. There was an increase in milk yield (from 13.96 to 16.71 kg/day) and increases in milk fat (quadratic, *p* = 0.04), protein (quadratic, *p* = 0.10), and lactose concentrations (linear, *p* = 0.07) with increasing APG supplementation. In addition, APG supplementation increased (*p* ≤ 0.03) the milk fat, protein, solid non-fat, and total solid yields, while the lactose yield increased (linear, *p* = 0.01) as the APG level increased. Dietary APG supplementation had no effect on nutrient digestibility and blood metabolites. It was concluded that the addition of APG at doses up to 22 mL/kg of pelleted concentrate had positive effects on the milk composition in dairy cows.

## 1. Introduction

Surfactants can increase the rate of hydrolysis of Sigmacell 100 cellulose and steam-exploded wood [1]. The enhancement of fibrolytic activity by surfactants can involve various mechanisms. In the cellulolytic ascomycete *Neurospora crassa*, Tween 80 stimulated the release of both endoglucanase and exoglucanase [2]. Kim et al. [3] also showed that surfactants can prevent the deactivation of cellulases, prolonging their lifespan.

In rumen cultures, surfactants have been found to enhance fibrolysis [4]. However, Hristov et al. [5] did not observe effects of Tween 80 on carboxymethyl cellulase or xylanase activities. Kim et al. [6] found that Tween 80 increased the total volatile fatty acid (VFA) concentration and carboxymethyl cellulose activity and increased or tended to increase the total tract digestibility of crude fibre (CF) and ether extract (EE) in Hanwoo steers.

Alkyl polyglycoside (APG) is a naturally produced, environmentally friendly, non-ionic surfactant (NIS) resulting from the reaction of glucose with long chain fatty alcohols [7]. Moreover, APG has also been utilized in detergents as well as drug delivery, food processing, and agrochemistry processes [8,9,10,11] owing to its low toxicity and good biological degradability. Our previous study showed that APG supplementation increased in vitro rumen gas production and dry matter (DM) and organic matter (OM) degradation of low-quality roughages [12]. Dietary APG supplementation increased OM intake and the digestibility of neutral detergent fibre (NDF) and acid detergent fibre (ADF) in goats [13]. Dietary APG supplementation affected the amino acid composition of the rumen content in goats [14]. To our knowledge, there is no information on the use of APG as additive supplement to dairy cow diets, even though data showed that poloxalene, a NIS, had no effect on the yield or composition of milk [15,16]. It is hypothesized that the APG supplementation could increase the nutrient digestibility, and then affect lactating performance of middle- and low-yielding dairy cows. The objective of this study was to characterize the responses of milk yield and composition, nutrient digestibility and blood metabolites to dietary APG supplementation in lactating dairy cows.

## 2. Materials and Methods

The Laboratory Animal Welfare and Animal Experimental Ethical Inspection Committee at the Institute of Subtropical Agriculture, Chinese Academy of Sciences reviewed and approved all protocols used in this study. The ethic code is No. ISA000185. The animal trials were conducted at the experimental farm of the Institute of Subtropical Agriculture (Changsha, China).

### 2.1. Animals Experimental Design and Diets

Twenty Holstein dairy cows, at 167 ± 69 days in milk, 480 ± 55 kg body weight and 14.1 ± 2.16 kg/day milk yield (mean ± SD), were randomly assigned to four groups (APG0, APG5.5, APG11, and APG22) that were supplemented with 4 levels of APG: 0, 5.5, 11 and 22 mL per pelleted concentrate (on a DM basis), respectively. The APG supplementation levels and methods referred to Yuan et al. [13]. The pre-experimental treatment adaptation time was 7 days, and the experiment lasted for 60 days. Cows were housed and fed individually in tie stalls with free access to clean water throughout the experiment. Each cow received 5 kg (4.5 kg DM) pelleted concentrate and 9 kg (2.1 kg DM) distillers grains daily, and a sufficient amount of fresh limpograss (*Hemarthria altissima*) was fed to each cow every day. The pelleted concentrate, distillers grains, and limpograss were fed separately, and equally and successively offered twice a day at 07:00 and 18:00. For each feeding, the distillers grains was offered after pelleted concentrate was eaten up, and then limpograss was fed after distillers grains was finished. The APG (appearing as a colourless to pale yellow liquid, with 8–10 carbon alkyl chains, 70% solid content and a hydrophilic-lyophilic balance of 13–15, provided by Jinmoer Chemical Co., Ltd., Shijiazhuang, China) was mixed by hand with pelleted concentrate before every feeding. This amount of pelleted concentrate and distillers grains were always completely eaten by each cow during the entire experimental phase, and fresh limpograss offered to and refused by an individual animal was recorded daily. The formula of the pelleted concentrate and chemical compositions of the diet are shown in Table 1.

### 2.2. Sample Collection and Processing

Cows were milked twice daily at 05:00 and l6:00. Individual milk yield was recorded daily. Milk samples were collected in a plastic bottle containing potassium dichromate at morning and afternoon milking sessions for two consecutive days every 20 days and combined according to the corresponding volume measured at each milking session. Milk samples were kept refrigerated (4 °C) and transported to the laboratory for milk composition analysis as soon as possible.

Blood samples were collected from the coccygeal vein of each cow before the afternoon milking on the 20th, 40th, and 60th days of the experimental period. Blood was collected into a vacuum serum tube (5 mL, Aosaite Medical Instrument Co., Ltd., Heze, China). Serum was separated by centrifugation at 3000 × g for 15 min and stored at −20 °C until analysis.

Feed samples were collected every 10 days, weighed and dried at 65 °C for 48 h, and then stored in a freezer until analysis. The amount of feed offered and refused was recorded while total faeces were collected daily for the last six consecutive days. The total collection of faeces at spontaneous defecation was performed using separate pans for each animal. The faeces in each container were weighed daily, and then a 10% aliquot of total faecal output was subsampled and frozen (−20 °C). The subsamples were later thawed and mixed to obtain one pooled sample from each animal. Each of these pooled samples was dried at 65 °C for 48 h and then analysed for estimation of digestibility.

### 2.3. Chemical Analyses

Milk samples were analysed for fat, protein, lactose, solid non-fat (SNF) and total solid (TS) contents (Foss Milko-Scan134A/B, Foss Electric, Hillerod, Denmark).

The serum concentrations of glucose (GLU), blood urea nitrogen (BUN), ammonia (AMM), β-hydroxybutyric acid (BHB), triglyceride (TG), cholesterol (TC), high-density lipoprotein cholesterol (HDL-C), and low-density lipoprotein cholesterol (LDL-C) concentrations were determined using an automatic biochemical analyser (Mindray BS-300, Mindray Medical International Limited, Shenzhen, China).

The standard method of the Association of Official Analytical Chemists (AOAC) [17] was followed to determine the proximate chemical compositions of diets and faeces. Ground samples of feed and faeces were analysed for DM by drying at 105 °C. Crude ash was determined by heating at 550 °C. Crude protein (CP = N × 6.25) was determined by the Kjeldahl method. EE was analysed using an automatic extraction apparatus (SOX416, C. Gerhardt GmbH & Co. KG, Nordrhein-Westfalen, Germany). NDF and ADF were determined according to the methods of [18] by using a Fibretherm FT 12 fibre analyser (C. Gerhardt GmbH & Co. KG, Nordrhein-Westfalen, Germany). Calcium in the basal diet was determined using an atomic absorption spectrophotometer (GBC, GBC932, Braeside VIC, Australia), and dietary phosphorus was determined by the colorimetric method.

### 2.4. Statistical Analyses

Dry matter intake (DMI) and milk production were calculated as an average per cow over the experimental period. Statistical analyses of DMI, milk production, and digestibility data were performed using the mixed procedure of SAS (1999) (SAS Institute Inc., Cary, NC, USA). Statistical analyses of milk composition were performed with repeated measures for sampling time referring to the mixed procedure of SAS [19] with a model including treatment, time, and time × treatment. Orthogonal polynomial contrasts were used to evaluate the linear and quadratic effects of APG supplementation levels. Significance was declared at *p* ≤ 0.05, and a tendency was declared at 0.05 ≤ *p* ≤ 0.10.

## 3. Results

As shown in Table 2, there was no significant difference in the DMI and milk yield with the addition of APG to diets. However, the milk yield in the APG supplement groups was numerically higher than that in the control. For milk composition, milk fat content had a quadratic response (*p* = 0.04), lactose content (*p* = 0.07) and lactose yield (*p* = 0.01) increased linearly, and protein content had an increasing quadratic trend (*p* = 0.10) with increasing APG levels. In addition, the milk protein yield, SNF yield and TS yield were significantly influenced by the addition of APG (*p* < 0.05), as they increased linearly (*p* = 0.03, *p* = 0.01, *p* = 0.01, respectively) as APG supplementation increased. The milk lactose yield increased linearly (*p* = 0.01) with increasing APG levels. Remarkably, milk fat yield was significantly associated (*p* = 0.03) with APG supplementation and exhibited quadratic growth (*p* = 0.01) with increasing levels of APG supplementation. The milk fat yield of the APG11 group was higher (*p* < 0.05) than that of the other groups. The lactose yield in APG22 group was significantly higher than that in APG0 group (*p* < 0.05). Cows in the APG11 and APG22 groups had greater (*p* < 0.05) milk protein, SNF and TS yields than cows in the APG0 group.

Blood index results in Table 3 show that BHB, AMM, TG, TC, LDL-C, and HDL-C in serum were not influenced by APG supplementation. However, in response to the increasing level of dietary APG supplementation in cows, the concentration of serum glucose showed a decreasing tendency (quadratic, *p* = 0.06).

In the present work, we also demonstrated that the apparent digestibility of CP, EE, crude ash, OM, NDF, and ADF (Table 4) in dietary diets was not influenced by APG supplementation.

## 4. Discussion

The research involving NIS in areas of livestock production has mainly focused on Tween 80, and there are a few studies on the application of APG and other NIS. In the present work, we found that the addition of APG in diets had no significant influence on the DMI of dairy cows, which is similar to the report of Kim et al. [20], who found no effect of dietary NIS (sorbitan trioleate) supplementation (50 mL/day/head) on the DMI of cows during the transition period. We estimate that due to the physical properties of APG, including good wettability, low interfacial tension [21,22], and temperature insensitivity [8], the palatability of the diet is not changed and the feed consumption is not affected in dairy cows.

Previously, Helmer et al. [15] and Bezeau et al. [16] found that the NIS poloxalene had no effect on milk yield. Kim et al. [20] demonstrated that average milk production of dairy cows in the first six weeks postpartum was not affected by the dietary supplementation of NIS (mainly sorbitan trioleate) compared with the control group. In the present study, although the average milk yield in the APG supplementation groups was numerically greater, there was no statistically significant difference in the average milk yield between the APG supplementation groups and the control group. The results appear to justify that supplementing APG in quantities up to 99 mL per day does not deleteriously affect dairy animals. Considering the relatively low number of animals used in this study, further studies are needed to explore the effect of APG on dairy cow performance.

In our study, APG supplementation in the diets increased milk fat and milk protein yields. This supported a study by Kim et al. [20], who found that NIS supplementation significantly increased the milk fat percentage by approximately 3.9% in cows. It has been speculated that increasing milk fat content by NIS supplementation might be related to the influences of NIS on rumen microflora [20] since rumen fermentation products, such as acetate (a primary milk fat precursor) and butyrate, are used by the cow for energy and by the mammary glands to produce milk fat [23]. Moreover, NIS have been shown to have an impact on rumen fermentation and the rumen microbial population [6,24,25,26,27]. On the other hand, our previous study indicated that the proportions of ruminal amino acids, such as valine, phenylalanine and lysine, were affected by the dietary APG supplementation in goats [14]. The amino acid profile in the rumen, especially the amino acid composition of microbial crude protein, have a close relationship with the protein and amino acid compositions of milk [28,29]. Therefore, the effects of dietary APG supplementation on milk fat and protein yield in the current study might be related to influencing the ruminal microbiota and amino acid composition. Unfortunately, we did not measure in vivo rumen fermentation and ruminal microbiota. However, this speculation contradicts the results that dietary APG inclusion did not affect nutrient digestibility in the study. So, the reason for how the APG affected the lactation performance is not clear, and further studies with a larger sample size and the measurement of fermentation characteristics and rumen microbiome need to be performed to shed more light on this occurrence.

For the blood index, there was no difference in BHB, AMM, BUN, TG, TC, LDL-C, and HDL-C in serum by dietary treatment, while the concentration of serum glucose showed a decreasing tendency as dietary APG supplementation increased. A previous study by our research group demonstrated that plasma urea and glucose concentrations in goats were not affected by dietary APG supplementation (13 mL daily per animal) [14]. Similarly, Hristov et al. [30] found that Tween 80 did not influence plasma glucose and urea concentrations in steers fed a diet containing 70% barley. In addition, Lee et al. [25] reported that the blood glucose values of dairy cows from 21 days prepartum through 42 days postpartum were also not affected by the addition of Tween 80.

For the apparent digestibility of CP, EE, crude ash, OM, NDF, and ADF, dietary APG supplementation had no significant influence, which is inconsistent with our hypothesis. In our prior study, dietary APG supplementation in goats increased the total tract digestibility of OM and NDF [13]. However, Chen et al. [31] reported that dietary supplementation with Tween 40, 60 or 80 (10 g/day, respectively) did not affect the nutrient intake or digestion of sheep. The dietary addition of Tween 80 (5 g/day or 10 g/day) had no effect on the digestibility of DM, CP, NDF, ADF, and crude ash in Hanwoo steers [6]. In addition, Hristov et al. [30] and McAllister et al. [32] reported that the apparent digestibility of DM, NDF, ADF, and CP were not affected by dietary supplementation with Tween 80 in steers and lambs. The reports on the effects of NIS on nutrient digestibility in animals are inconsistent, although most of the existing studies showed that NIS had no effect on digestibility. The differences in the results of NIS on nutrient digestibility among studies were probably due to the experimental animals, experimental diets, NIS inclusion levels, NIS properties, etc. Thus, more studies are needed to investigate the effect of NIS on nutrient digestibility.

## 5. Conclusions

Dietary APG supplementation improved milk fat, milk protein, SNF and TS outputs in lactating dairy cows; no negative responses in milk production, digestibility, and blood metabolites were observed. This implies that APG has potential as a feed additives to improve lactation performance in middle and low yielding dairy cattle. Further studies are needed to explore the mode of action of APG in dairy cows, as well as the influence of APG on ruminal fermentation, microbial communities, and interfacial nutrient metabolism of ruminal microbes.

## Figures and Tables

**Table 1 animals-09-00549-t001:** Ingredients and chemical compositions of the experimental diets.

Composition (% DM) ^1^	Concentrate	Distillers Grains	Forage Grass
Ingredients			
Corn	52.5		
Wheat bran	11.9		
Soybean meal	16.2		
Cotton meal	8.2		
Rapeseed meal	5.0		
Calcium carbonate	1.7		
Calcium hydrophosphate	1.1		
Sodium chloride	1.1		
Mineral and vitamin mix ^2^	2.3		
Distillers grains		100	
Limpograss			100
Chemical compositions			
DM	90.4	23.0	25.5
CP	22.9	39.8	11.5
EE	2.97	12.9	2.96
Crude ash	6.77	3.74	8.72
NDF	27.7	37.5	68.5
ADF	14.4	14.9	37.3
Ca	0.78	0.05	0.20
P	0.46	0.15	0.11

^1^ DM = dry matter; CP = crude protein; EE = ether extract; NDF = neutral detergent fibre; ADF = acid detergent fibre; Ca = calcium; P = phosphorus. ^2^ Mineral and vitamin premix contained 3170 mg/kg Fe, 14,280 mg/kg Zn, 3060 mg/kg Mn, 3040 mg/kg Cu, 40 mg/kg Co, 180 mg/kg I, 100 mg/kg Se, 1,250,000 IU/kg vitamin A, 270,000 IU/kg vitamin D, and 5,000 IU/kg vitamin E.

**Table 2 animals-09-00549-t002:** Effects of APG supplementation on the lactating performance of cows.

Item ^1^	Treatment ^2^				SEM	*p*-Value ^3^		
	APG0	APG5.5	APG11	APG22		T	L	Q
DMI ^4^	11.0	11.3	11.1	11.2	0.14	0.65	0.60	0.71
Milk yield	14.0	15.2	16.3	16.7	1.17	0.42	0.14	0.50
Yield (g/d)								
Fat	460.6 ^b^	540.7 ^ab^	649.5 ^a^	547.5 ^ab^	38.55	0.03	0.13	0.01
Protein	412.8 ^c^	465.5 ^bc^	560.0 ^a^	513.1 ^ab^	30.76	0.02	0.03	0.03
Lactose	653.4 ^b^	706.4 ^ab^	767.6 ^ab^	819.0 ^a^	41.23	0.06	0.01	0.57
SNF	1216.5 ^b^	1333.3 ^ab^	1504.2 ^a^	1524.3 ^a^	72.31	0.02	0.01	0.19
TS	1636.6 ^b^	1825.0 ^ab^	2091.0 ^a^	2013.6 ^a^	97.41	0.02	0.01	0.05
Composition (%)								
Fat	3.29	3.58	4.02	3.29	0.26	0.19	0.96	0.04
Protein	2.96	3.07	3.46	3.06	0.18	0.25	0.62	0.10
Lactose	4.67	4.65	4.70	4.89	0.09	0.25	0.07	0.40
SNF	8.71	8.78	9.25	9.10	0.24	0.34	0.18	0.37
TS	11.7	12.0	12.9	12.0	0.43	0.29	0.54	0.11

^1^ DMI = dry matter intake; SNF = solid non-fat; TS = total solid. Data of milk composition and component yields shown are the mean of values on d 20, 40 and 60. Means within a row with different superscripts are different at *p* < 0.05. ^2^ APG = alkyl polyglycoside; APG0, APG5.5, APG11, APG22 = add 0, 5.5, 11 and 22 mL APG per kg pelleted concentrate on a DM basis, respectively. ^3^ T = treatment; L = linear; Q = quadratic. Means with different letters within a row differ significantly (*p* < 0.05). ^4^ DMI = pelleted concentrate + distillers grains + limpograss. The pelleted concentrate, distillers grains, and limpograss were fed separately. Each dairy cow was fed equal amounts of pelleted concentrate (4.5 kg DM/cow) and distillers grains (2.1 kg kg DM/cow) daily; A sufficient amount of limpograss was fed to each cow every day and the intake of limpograss was recorded.

**Table 3 animals-09-00549-t003:** Effects of APG supplementation on the blood index of lactating cows.

Item (mmol/L) ^1^	Treatment ^2^				SEM	*p*-Value ^3^		
	APG0	T	APG11	APG22		T	L	Q
GLU	1.94	1.75	1.49	1.82	0.15	0.21	0.59	0.06
BHB	0.29	0.29	0.29	0.30	0.02	0.93	0.63	0.71
AMM	0.06	0.06	0.59	0.06	3.21	0.96	0.88	0.95
BUN	3.83	3.49	3.65	3.96	0.24	0.53	0.49	0.26
TG	0.91	1.27	1.31	2.41	0.68	0.46	0.14	0.74
TC	2.44	2.35	1.92	2.36	0.32	0.62	0.84	0.29
LDL-C	2226.7	2028.7	1721.1	2142.6	292.6	0.64	0.85	0.24
HDL-C	2.13	2.10	1.87	2.15	0.22	0.76	0.99	0.40

^1^ GLU = glucose; BHB = beta-hydroxybutyrate; AMM = ammonia; BUN = urea nitrogen; TG = triglyceride; TC = total cholesterol; LDL-C = low-density lipoprotein cholesterol; HDL-C = high-density lipoprotein cholesterol. ^2^ APG = alkyl polyglycoside; APG0, APG5.5, APG11, APG22 = 0, 5.5, 11 and 22 mL APG per kg of pelleted concentrate on a DM basis, respectively. ^3^ T = treatment; L = linear; Q = quadratic.

**Table 4 animals-09-00549-t004:** Effects of APG supplementation on digestibility in lactating cows.

Item (%) ^1^	Treatment ^2^				SEM	*p*-Value ^3^		
	APG0	APG5.5	APG11	APG22		T	L	Q
CP	74.3	71.1	74.8	74.7	2.43	0.67	0.65	0.71
EE	82.6	81.3	82.1	83.1	2.18	0.94	0.76	0.67
Crude ash	41.3	38.1	43.1	38.3	5.41	0.88	0.81	0.81
OM	68.6	64.1	68.5	67.1	2.94	0.69	0.98	0.76
NDF	59.3	53.6	59.1	55.8	3.92	0.71	0.77	0.89
ADF	45.8	41.5	49.2	44.6	4.44	0.65	0.92	0.78

^1^ CP = crude protein; EE = ether extract; OM = organic matter; NDF = neutral detergent fibre; ADF = acid detergent fibre. ^2^ APG = alkyl polyglycoside; APG0, APG5.5, APG11, APG22 = 0, 5.5, 11 and 22 mL APG per kg of pelleted concentrate on a DM basis, respectively. ^3^ T = treatment; L = linear; Q = quadratic.

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
