# Peer review of "Effects of Dietary Alkyl Polyglycoside Supplementation on Lactation Performance, Blood Parameters and Nutrient Digestibility in Dairy Cows"

_animals, 2019, doi:10.3390/ani9080549_

Round 1
Reviewer 1 Report
Review of manuscript number: animals-544909
Title: Effects of dietary alkyl polyglycoside supplementation on lactation performance, blood parameters and nutrient digestibility in dairy cows.
The purpose of the research was to determine the effect of dietary alkyl polyglycoside (APG) supplementation for lactating dairy cows on milk yield and composition, nutrient digestibility and blood metabolites.
The study was performed on 20 cows. They were divided into four groups, depending on the amount of APG administered: APG0, APG5.5, APG11, and APG22, that were supplemented with 4 levels of APG: 0, 5.5, 11 and 22 mL per pelleted concentrate (on a dry matter), respectively.
In all cows, the milk yield and milk composition, biochemical blood tests and chemical composition of diet and feces were evaluated. Milk and blood were collected every 20 days and feed and feces every 10 days. I consider that doing this research required a lot of work and commitment from the authors.
In general, the work is written correctly. The study was well planned and performed. Some of the results are also interesting.
The paper shows that the use of alkyl polyglycoside (APG) in feed (mainly APG11, APG22), causes an increase in milk yield and increase in fat, protein, lactose, solid non-fat, total solid in milk. APG does not improve feed digestibility
Currently, dairy cows have too high milk yield, which can lead to the development of many diseases, including ketosis, acidosis, fatty liver and infertility. As a consequence, they may be the reason for elimination of animals from the herd and large economic losses.
Is it necessary to use nutritional supplements to increase milk yield, in cows with high milk yield? Especially when the improvement of feed digestibility has not been demonstrated.
As for the individual parts of the manuscript: Introduction, Material and methods, Results, Discussion, these parts were written correctly, I did not notice any obvious (serious) errors.
During the detailed analysis of the text, some minor irregularities were noticed. I hope that improving them will increase the value of the manuscript.
At the beginning of each part of the manuscript, the abbreviations used should be explained.
Introduction
At the end of this section should clearly define the purpose of the study.
Tables
In my opinion, the tables are not legible. The reader must be well-versed how to link the results to the statistics. e.g. blood tests were performed 3 times every 20 days in each group, but the result is only one. e.g. GLU in group APG22 is 1.82, from which day of the study it is a result, from 20th, 40th or 60th. Similarly, other results.
I do not think that new tables should be made. However, I think that they should be better described or explained in the text.
Discussion
L 186 The authors should try to explain why there is an increase in milk yield and growth of the tested milk components without improving feed digestibility.
L 192 Please add how impact has NIS on rumen fermentation and the rumen microbial population. It is currently that NIS has an impact, but how it affects…
I like the assumptions made in the conclusions; “Further studies are needed to explore the mode of action of APG in dairy cows, as well as the influence of APG on ruminal fermentation, microbial communities, and interfacial nutrient metabolism of ruminal microbes.” In my opinion, this is a good direction for further research, which may explain the results of the research described in the reviewed manuscript
I do not want to evaluate this work negatively. But I know that more modern work is being removed from the Animals.
Only from the editor in chief of the Animals depends on whether this manuscript will be accepted for publication in this journal.
Author Response
Response to Reviewer Comments
Dear Reviewer,
Thank you for your comments and suggestions. Those comments and suggestions are all valuable and helpful for revising and improving our manuscript, as well as guiding our researches. We have studied comments carefully and have made revisions according to your comments and suggestions, as described below.
Point 1: The purpose of the research was to determine the effect of dietary alkyl polyglycoside (APG) supplementation for lactating dairy cows on milk yield and composition, nutrient digestibility and blood metabolites.
The study was performed on 20 cows. They were divided into four groups, depending on the amount of APG administered: APG0, APG5.5, APG11, and APG22, that were supplemented with 4 levels of APG: 0, 5.5, 11 and 22 mL per pelleted concentrate (on a dry matter), respectively.
In all cows, the milk yield and milk composition, biochemical blood tests and chemical composition of diet and feces were evaluated. Milk and blood were collected every 20 days and feed and feces every 10 days. I consider that doing this research required a lot of work and commitment from the authors.
In general, the work is written correctly. The study was well planned and performed. Some of the results are also interesting.
The paper shows that the use of alkyl polyglycoside (APG) in feed (mainly APG11, APG22), causes an increase in milk yield and increase in fat, protein, lactose, solid non-fat, total solid in milk. APG does not improve feed digestibility.
Currently, dairy cows have too high milk yield, which can lead to the development of many diseases, including ketosis, acidosis, fatty liver and infertility. As a consequence, they may be the reason for elimination of animals from the herd and large economic losses.
Is it necessary to use nutritional supplements to increase milk yield, in cows with high milk yield? Especially when the improvement of feed digestibility has not been demonstrated.
Response 1: Thanks a lot for your positive comments on our study. We fully agree with your views and comments on high-yielding cows. High producing dairy cows are susceptible to metabolic diseases. The need to improve the milk yield of dairy cows through nutritional strategies should depend on the cow production level, feeding pattern, feed situation, etc. In China and other countries or regions, there are many dairy farms where the cow populations which have middle and low yields, and local feed resources are used to feed the cows. For such farming model, there is still a need to use nutritional strategies to improve feed utilization and production performance of dairy cows. Therefore, according to your comments and suggestions, we have modified the description in the Introduction and Conclusion sections to indicate that the potential application targets are mainly middle- and low-yielding dairy cows.
Point 2: As for the individual parts of the manuscript: Introduction, Material and methods, Results, Discussion, these parts were written correctly, I did not notice any obvious (serious) errors.
During the detailed analysis of the text, some minor irregularities were noticed. I hope that improving them will increase the value of the manuscript.
Response 2: Thanks for your positive comments of our manuscript and we are sorry for the some irregularities in the manuscript. We have modified the manuscript according to your suggestions. All the revisions have been highlighted using red fonts throughout the revised manuscript.
Point 3: At the beginning of each part of the manuscript, the abbreviations used should be explained.
Response 3: A list of abbreviations has been prepared as suggested. Please see lines 40-46 in the revised version.
Point 4: Introduction: At the end of this section should clearly define the purpose of the study.
Response 4: Thanks for your precious suggestion. We have added the hypothesis and defined the purpose. Please see Lines 68-71.
Point 5: Tables: In my opinion, the tables are not legible. The reader must be well-versed how to link the results to the statistics. e.g. blood tests were performed 3 times every 20 days in each group, but the result is only one. e.g. GLU in group APG22 is 1.82, from which day of the study it is a result, from 20th, 40th or 60th. Similarly, other results.
I do not think that new tables should be made. However, I think that they should be better described or explained in the text.
Response 5: We are sorry to make you confused. We have revised the table according to your suggestion. Please see the Table 2 in the revised version.
Point 6: L 186 The authors should try to explain why there is an increase in milk yield and growth of the tested milk components without improving feed digestibility. L 192 Please add how impact has NIS on rumen fermentation and the rumen microbial population. It is currently that NIS has an impact, but how it affects…
Response 6: Thanks for your valuable advice. For the reason why there is an increase in milk production and milk components yields, one possibility is APG would affect rumen fermentation and gastrointestinal microbiota, however this is a speculation based on the literature. Unfortunately, we did not measure in vivo rumen fermentation and ruminal microbiota. Moreover, dietary APG inclusion did not improve the nutrient digestibility, which is inconsistent with this speculation and our original hypothesis. Therefore, it is unclear by what mechanism the APG affected the lactation performance. According to your comments and suggestions, we modified the description of the discussion section to make these descriptions more based on the analysis of the relative literature and the reasonable inference of the results in the current study. New animal experiments with larger sample size and more parameters to explore the mode of action of APGs on dairy cows need to be conducted in our future work. We are very sorry for this, and we sincerely hope your understanding.
Point 7: I like the assumptions made in the conclusions; “Further studies are needed to explore the mode of action of APG in dairy cows, as well as the influence of APG on ruminal fermentation, microbial communities, and interfacial nutrient metabolism of ruminal microbes.” In my opinion, this is a good direction for further research, which may explain the results of the research described in the reviewed manuscript
Response 7: Thanks for your positive comments and precious suggestion.

Reviewer 2 Report
General comments:
1. This study is interesting since there is absence of information on the use of APG as additive supplement to dairy cow diets.
2. However, my concern is that the study is dealing with local cows eating around 11 kg DM/d and producing only 14 to 16.7 kg milk/d. This DMI level is around 40% of the average DMI of Holstein cows (25 kg/d) that produce above 35 kg milk/d. This gap between the cows used in the current study and the modern cows used in developed countries, reduces the importance of this study for the global community.
3. In view of comment 2, the conclusion of the authors (L. 225) that: "This implies that APG has potential as a feed additives to improve lactation performance in dairy cattle", is very speculative and should be restricted only to low producing local cows.
4. The authors should provide in Table 1 the ingredients of the TMR (g/kg DM TMR) since the proportion of concentrated pellets, DG and forage in the TMR is not clear. The ingredients of the pelleted concentrates should be provide in the legends or alternatively as g/kg DM TMR within the table. Moreover, chemical composition data of the entire TMR should be given in addition to composition data of concentrates, distillers grains and forage.
5. In Tables 2, 3 and 4 mean values in the same raw (dietary treatments) that differ significantly from each other at p<0.05 should be marked with different superscripts. For example, it is not clear if APG11 is better than APG22 in milk solids yield and composition.
6. In the discussion authors tried to explain the mechanism of APG effect on fat yield by speculating that: "dietary APG supplementation affected milk fat and protein yield possibly by influencing the ruminal microbiota and amino acid composition". This speculation is not base on any measurement of pH or VFA concentrations in the rumen or any count of rumen cellulolytic microbiota and therefore is not valid. Moreover, any explanation of this kind is in contrast to the finding that: NDF digestibility was not affected by the dietary treatments (Table 4).
Author Response
Response to Reviewer Comments
Dear Reviewer,
Thank you for your comments and suggestions. Those comments and suggestions are all valuable and helpful for revising and improving our manuscript, as well as guiding our researches. We have studied comments carefully and have made revisions according to your comments and suggestions, as described below.
Point 1: This study is interesting since there is absence of information on the use of APG as additive supplement to dairy cow diets.
Response 1: Thanks for your positive comments of our manuscript.
Point 2: However, my concern is that the study is dealing with local cows eating around 11 kg DM/d and producing only 14 to 16.7 kg milk/d. This DMI level is around 40% of the average DMI of Holstein cows (25 kg/d) that produce above 35 kg milk/d. This gap between the cows used in the current study and the modern cows used in developed countries, reduces the importance of this study for the global community.
Response 2: Thanks for your comments. The cows in our study were middle yielding cows in the south of China and these cows were during middle and late lactation stages. In the Hunan province (southern central China) where this study was conducted, the cows here are middle or low yielding cows, which is related with cows population, climate (for example, high temperature and humidity in Summer), local feed resources, and feeding pattern of dairy cows, etc. We agree with you that the results of our current study might provide a reference for the production of middle and low yielding dairy cows.
Point 3: In view of comment 2, the conclusion of the authors (L. 225) that: "This implies that APG has potential as a feed additives to improve lactation performance in dairy cattle", is very speculative and should be restricted only to low producing local cows.
Response 3: Thanks for your precious comments. We have revised the manuscript according your suggestion. Please see Line 248.
Point 4: The authors should provide in Table 1 the ingredients of the TMR (g/kg DM TMR) since the proportion of concentrated pellets, DG and forage in the TMR is not clear. The ingredients of the pelleted concentrates should be provide in the legends or alternatively as g/kg DM TMR within the table. Moreover, chemical composition data of the entire TMR should be given in addition to composition data of concentrates, distillers grains and forage.
Response 4: Thanks for your comments. The pelleted concentrate, distillers grains and fresh limpograss were fed separately. We are sorry that we did not described it clearly. We have added the descriptions in the revised version. Please see Lines 85-88.
Point 5: In Tables 2, 3 and 4 mean values in the same raw (dietary treatments) that differ significantly from each other at p<0.05 should be marked with different superscripts. For example, it is not clear if APG11 is better than APG22 in milk solids yield and composition.
Response 5: We have added the marks in Table 2 and the corresponding description according to your suggestion. Please see the Table 2 and in Lines 150-156 in the revised version.
Point 6: In the discussion authors tried to explain the mechanism of APG effect on fat yield by speculating that: "dietary APG supplementation affected milk fat and protein yield possibly by influencing the ruminal microbiota and amino acid composition". This speculation is not based on any measurement of pH or VFA concentrations in the rumen or any count of rumen cellulolytic microbiota and therefore is not valid. Moreover, any explanation of this kind is in contrast to the finding that: NDF digestibility was not affected by the dietary treatments (Table 4).
Response 6: Thanks for your comments!We have modified the description in the discussion section according to your comments and suggestions in the revised version.

Round 2
Reviewer 2 Report
Most of the changes were appropriately made according to my previous suggestions.
However, in Table 2 DMI should be separated into 3 raws: DMI from pellets+ AP, DMI from DG and DMI from hay.
Author Response
Response to Reviewer Comments Dear Reviewer, Thank you for your comments and suggestions. Those comments and suggestions are all valuable and helpful for revising and improving our manuscript, as well as guiding our researches. We have studied comments carefully and have made revisions according to your comments and suggestions, as described below. Point 1: Most of the changes were appropriately made according to my previous suggestions. However, in Table 2 DMI should be separated into 3 raws: DMI from pellets+ AP, DMI from DG and DMI from hay. Response 1: Thanks a lot for your positive comments to our modifications made according to your previous comments and suggestions. About your suggestions on DMI in Table 2, we thank and agree with you. At the same time, taking into account the actual situation of our study, ie, “Each cow received 5 kg (4.5 kg DM) pelleted concentrate and 9 kg (2.1 kg DM) distillers grains daily, and a sufficient amount of fresh limpograss (Hemarthria altissima) was fed to each cow every day.” We made the following modifications: According to your suggestions and in combination with the above actual situation, we have modified the corresponding description in the Materials and Methods section, and added a table note description, ie, “DMI = pelleted concentrate + distillers grains + limpograss. The pelleted concentrate, distillers grains, and limpograss were fed separately. Each dairy cow was fed equal amounts of pelleted concentrate (4.5 kg DM/cow) and distillers grains (2.1 kg kg DM/cow) daily; A sufficient amount of limpograss was fed to each cow every day and the intake of limpograss was recorded.” We hope that such changes will meet your suggestions and look forward to your recognition.
